# Impacts of snow data and processing methods on the interpretation of long-term changes in Baffin Bay early spring sea ice thickness

Isolde A Glissenaar[1], Jack C Landy[1,2], Alek A Petty[3,4], Nathan T Kurtz[3], and Julienne C Stroeve[5,6,7]

[1]Bristol Glaciology Centre, School of Geographical Sciences, University of Bristol, Bristol, UK
[2]The Earth Observation Laboratory, Department of Physics and Technology, UiT The Arctic University of Norway, Tromsø, Norway
[3]Cryospheric Sciences Laboratory, NASA Goddard Space Flight Center, Greenbelt, MD, USA
[4]Earth System Science Interdisciplinary Center, University of Maryland, College Park, MD, USA
[5]Centre for Earth Observation Science, University of Manitoba, Winnipeg, MB, Canada
[6]Department of Earth Science, University College London, London, United Kingdom
[7]National Snow and Ice Data Center, University of Colorado Boulder, Boulder, CO, USA

**Correspondence:** Isolde Glissenaar (isolde.glissenaar@bristol.ac.uk)

**Abstract.** In the Arctic, multi-year sea ice is being rapidly replaced by seasonal sea ice. Baffin Bay, situated between Greenland and Canada, is part of the seasonal ice zone. In this study, we present a long-term multi-mission assessment (2003-2020) of spring sea ice thickness in Baffin Bay from satellite altimetry and sea ice charts. Sea ice thickness within Baffin Bay is calculated from Envisat, ICESat, CryoSat-2 and ICESat-2 freeboard estimates, alongside a proxy from the ice chart stage of development that closely matches the altimetry data. We study the sensitivity of sea ice thickness results estimated from an array of different snow depth and snow density products and methods for redistributing low resolution snow data onto along-track altimetry freeboards. The snow depth products that are applied include a reference estimated from the Warren climatology, a passive microwave snow depth product, and the dynamic snow scheme SnowModel-LG. We find that applying snow depth redistribution to represent small-scale snow variability has a considerable impact on ice thickness calculations from laser freeboards but was unnecessary for radar freeboards. Decisions on which snow loading product to use and whether to apply snow redistribution can lead to different conclusions on trends and physical mechanisms. For instance, we find an uncertainty envelope around the March mean sea ice thickness of 13% for different snow depth/density products and redistribution methods. Consequently, trends in March sea ice thickness from 2003-2020 range from -23 cm/dec to 17 cm/dec, depending on which snow depth/density product and redistribution method is applied. Over a longer timescale, since 1996, the proxy ice chart thickness product demonstrates statistically significant thinning within Baffin Bay of 7 cm/dec. Our study provides further evidence for long-term asymmetrical trends in Baffin Bay sea ice thickness (with -17.6 cm/dec thinning in the west and 10.8 cm/dec thickening in the east of the bay) since 2003. This asymmetrical thinning is consistent for all combinations of snow product and processing method, but it is unclear what may have driven these changes.

# 1 Introduction

Arctic sea ice concentration and thickness has reduced significantly in recent decades (Kwok, 2018; Stroeve and Notz, 2018). With a >50% decrease in multi-year ice (MYI) cover since the turn of the century, the Arctic is increasingly becoming dominated by seasonal ice (Kwok, 2018). The Arctic winter area coverage of seasonal ice has surpassed that of MYI, making the understanding of processes over first-year ice (FYI) to be as important as those over MYI (Jeffries et al., 2013).

Regions in the Arctic where the perennial ice (sea ice that remains at the end of summer) has declined rapidly include the Chukchi and East Siberian Seas, where there is nowadays virtually no perennial ice left (Stroeve and Notz, 2018). The Kara Sea, Barents Sea and Baffin Bay are generally composed of seasonal ice. It is important to understand the interannual variation in the seasonal ice zone as these are generally regions of current and future shipping activity where thickness estimations are critical for safety (Christensen et al., 2018) and FYI regions have a strong influence on summer ice extent forecasting accuracy (Day et al., 2014; Serreze and Stroeve, 2015).

Most of the recent efforts to produce sea ice thickness estimates from satellite altimetry have focused on the central Arctic sea ice pack which is dominated by MYI (Kwok and Cunningham, 2015; Laxon et al., 2013; Petty et al., 2020; Ricker et al., 2017; Tilling et al., 2018). This is largely due to increased density of in-situ snow measurements in the more central Arctic (Warren et al., 1999) and issues with waves/freeboard determination around the sea ice edge. There are thus generally fewer published records focused on providing and estimating ice thickness and trends in the peripheral seas (Mallett et al., 2020). It is becoming particularly important to create a reliable sea ice thickness product for the seasonal ice zone with the ongoing rapid replacement of MYI by seasonal ice.

Baffin Bay, situated between Greenland and the Canadian Arctic (Fig. 1), is part of the seasonal ice zone. There is an import of multi-year ice from the Arctic Ocean through Nares Strait, but most of Baffin Bay is covered with first-year ice and the entire bay is ice free in summer. Baffin Bay plays a key role in modulating the freshwater flux from the Arctic basin to the Labrador Sea, which is a key location of North Atlantic Deep-Water formation, the deep-water component of the Atlantic Meridional Overturning Circulation (Cuny et al., 2005; Curry et al., 2011; Fenty and Heimbach, 2013; Holland et al., 2001). Moreover, Baffin Bay is a busy shipping region (Christensen et al., 2018) and an important area for polar bear migration (Obbard et al., 2010).

Previous research has suggested an apparent asymmetry in Baffin Bay spring sea ice thickness, with a thicker ice pack in the west of the bay than in the east (Landy et al., 2017). Landy et al. (2017) also demonstrated possible long-term asymmetrical trends in sea ice thinning, with stronger thinning in the western part of Baffin Bay over the ICESat-CryoSat-2 period (2003-2016).

Radar and laser altimetry data can be used to calculate sea ice freeboard and estimate sea ice thickness. The altimetry approach measures the height of the snow or ice surface elevation together with local sea level, which can be used to determine the freeboard – the extension of ice above sea level. To determine sea ice thickness from sea ice freeboard data, information is needed on the depth of the snowpack on the ice, the snow density, the sea ice density, and the ocean water density.

Uncertainty in snow depth is one of the largest error sources in sea ice thickness determination from satellite altimetry, contributing up to 50% of the total radar altimetry thickness uncertainty (Giles et al., 2007) and potentially 70% of laser altimetry thickness uncertainty (Zygmuntowska et al., 2014). Multiple snow depth products are available to retrieve sea ice thickness from satellite altimetry (Zhou et al., 2021). Past studies have generally used the estimation of snow depth across the Arctic from the Warren climatology (Warren et al., 1999). This climatology is based on in-situ snow depth observations collected between 1954-1991 in a limited region in the Arctic Ocean. Another basin-scale snow depth approach was developed by Markus and Cavalieri (1998) and relies on satellite passive microwave (PMW) brightness temperatures to retrieve snow depth over thin ice. This approach was thought to be valid only over seasonal ice, however more recent work has produced similar PMW snow estimates over multi-year ice (e.g., Rostosky et al., 2018). Because of the uncertainties associated with the above two approaches, recent work has focused on modelling snow accumulation using atmospheric reanalysis data together with observations of sea ice drift and concentration (Blanchard-Wrigglesworth et al., 2018; Liston et al., 2020; Kwok and Cunningham, 2008; Petty et al., 2020). Lastly, snow depth was recently estimated by combining radar and laser altimetry data because the two sensors theoretically measure elevations of the snow-ice and air-snow interfaces, respectively (Kwok et al., 2020).

All current snow depth products typically have a much lower horizontal resolution (∼25-100 km) than the along-track freeboard measurements they are applied to (∼10-100 m for laser altimetry, ∼300-2000 m for radar altimetry). Previous studies have employed redistribution functions to represent small-scale variability not captured in the large-scale snow depth products such as new ice formation in leads and wind redistribution (Kurtz et al., 2009; Kwok and Cunningham, 2008; Petty et al., 2020). Kwok and Cunningham (2008) utilized a sigmoidal function of the large-scale snow depth and high-resolution ICESat freeboard measurements. When the total freeboard is close to, or less than, the large-scale snow depth, the effective local snow depth is taken to be a fraction of the total freeboard as defined by a sigmoidal function. More recently, Petty et al. (2020) applied an updated version of Kurtz et al. (2009) on ICESat-2 measurements using an empirical approach of fitting a piecewise function that was determined based on airborne measurements from the NASA Operation IceBridge campaigns. Snow depth redistribution schemes have been applied to laser altimetry data, i.e. ICESat and ICESat-2, but have not yet been tested for radar altimetry data.

In this study, we aim to reconcile the spring sea ice thickness derived from multiple satellite altimetry sensors and sea ice charts in Baffin Bay and produce a robust long-term record (2003-2020). We retrieve sea ice thickness from Envisat (2003-2012), ICESat (2003-2009), CryoSat-2 (2011-2020) and ICESat-2 (2019-2020) to look at the regional long-term sea ice thickness change. We use this long-term record to analyse possible asymmetrical trends in sea ice thinning in Baffin Bay. We investigate the impact of different snow depth and density products and redistribution methods on retrieved sea ice thickness from satellite altimetry along-track sea ice freeboard data. Moreover, we determine whether snow redistribution schemes need to be applied to radar freeboard data, as well as laser freeboard data, for accurate determination of the sea ice thickness.

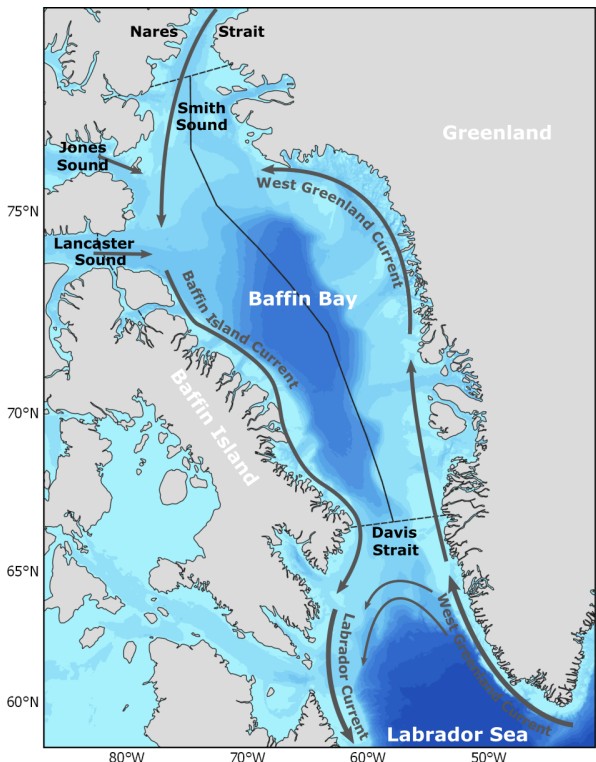

**Figure 1.** Schematic of the ocean currents and bathymetry of Baffin Bay separated in sections Western Baffin Bay and Eastern Baffin Bay. Bathymetry data from the GEBCO grid (GEBCO Compilation Group, 2019).

## 2 Methods

### 2.1 Freeboard observations from satellite altimetry

We use freeboard data from the following satellite altimetry missions to determine March sea ice thickness: ICESat (2003-2009), Envisat (2003-2012), CryoSat-2 (2011-2020) and ICESat-2 (2019-2020) (Fig. 2). We limit our study to March only to focus on end-of-winter/early spring sea ice thickness when Baffin Bay is fully ice covered. Also, the ICESat mission only produced data for specific quasi-monthly campaign periods, including February-March. The freeboard observations from the various satellite sensors used in this study are described below.

#### 2.1.1 ICESat

ICESat was launched in January 2003 by the National Aeronautics and Space Administration (NASA). The satellite orbits up to a latitudinal limit of 86° with a 91-day repeat period. The Geoscience and Laser Altimeter System (GLAS) on board ICESat provided surface elevation estimates within specific fall and spring campaign periods. The laser altimeter (wavelength 1064 nm had a laser footprint of ∼70 m in diameter spaced at ∼170 m intervals (Kwok et al., 2006). The laser stopped working in 2009.

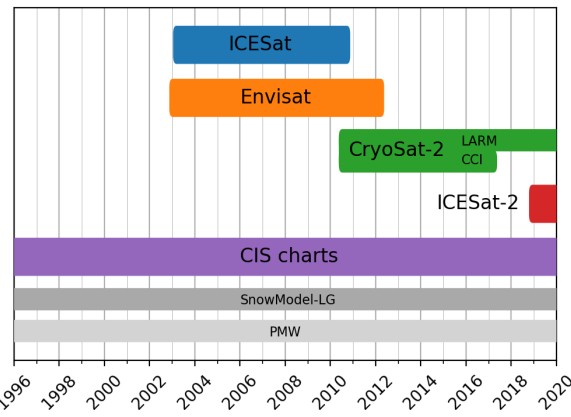

**Figure 2.** Timeline of different datasets used in this study.

Surface elevation relative to the TOPEX/Poseidon ellipsoid is provided by the National Snow and Ice Data Center (NSIDC) as the GLA13 product (Version 34, available at http://nsidc.org/data/icesat/ (Zwally et al., 2014)).

For the present study, elevation returns are filtered and corrected for geodetic and oceanographic biases, including geoid undulations, tides, dynamic topography of the ocean and the inverted barometer effects following Landy et al. (2017). We used elevation data where the sea ice concentration in the daily 25 km OSI-SAF global ice concentration reprocessing dataset (Andersen et al., 2012) was above 20%. For days with more than 100 elevation samples within the Baffin Bay/Hudson Bay area, returns outside 4 times the standard deviation in relative elevation are considered outliers and are removed. Sea ice (floe) and sea surface (lead) elevation samples were separated based on differences in the reflective properties and relative elevation of these surface types using the approach in Landy et al. (2017), applying an adapted version of the algorithms from Kwok et al. (2007) and Kwok and Cunningham (2008). Leads are identified where the reflectivity is more than 0.2 units below the background level in 25 km segments. A snow depth correction is applied to the elevation of the lead measurements (Kwok and Cunningham, 2008) and outlying leads are removed where the elevation is 3 times the standard deviation above the median or 2 times the standard deviation below the median. The sea level is interpolated between lead elevation measurements using cubic interpolation and smoothed using a 100 km window.

Total (snow plus sea ice) freeboard was calculated as the height difference between the ice surface elevation and the interpolated sea level measurements. Negative total freeboard measurements were removed. Freeboard uncertainty depends on the distance to the closest lead elevation sample. Freeboard uncertainty for sea ice measurements within 25 km from the closest lead sample were determined from uncertainty in sea surface height obtained from a 25 km running standard deviation of lead measurements (Landy et al., 2017). For measurements located further than 100 km away from the closest lead, the freeboard uncertainty is constrained only by the height difference between the ICESat observation and the sea surface height. For measurements between 25-100 km from the closest lead sample the freeboard uncertainty increases linearly with distance between these values.

### 2.1.2 Envisat

Envisat was launched in March 2002 by the European Space Agency (ESA). The satellite orbits up to a latitudinal limit of 81.5° with an orbit period of 100 minutes with a repeat cycle of 35 days. The intertrack spacing at the Equator is 80 km. The RA-2 altimeter on board Envisat includes a Ku-band pulse-limited radar altimeter (320 MHz), which theoretically penetrates snow if it is present on sea ice and measures sea ice elevation. The RA-2 altimeter has a nominal footprint of ∼2km (Connor et al., 2009). Communication with Envisat was lost in April 2012.

Envisat radar freeboard data on the satellite swath for March 2003-2012 were collected from ESA Climate Change Initiative (http://cci.esa.int/seaice) (Hendricks et al., 2018b). The surface-type classification is based on backscatter, leading-edge width, and pulse peakiness with the thresholds being determined with a combination of unsupervised clustering and supervised classification of waveforms from the central Arctic (Hendricks et al., 2018b; Paul et al., 2018). Retracking of the sea ice waveforms was done by applying a threshold first-maximum retracker algorithm where the threshold was calibrated to CryoSat-2 radar freeboard results in the Central Arctic. The product includes a radar freeboard uncertainty estimate.

### 2.1.3 CryoSat-2

CryoSat-2 (CS2) was launched in April 2010 by ESA. The satellite orbits up to a latitudinal limit of 88° on a 369-day repeat period with a 30-day subcycle. The intertrack spacing at the Equator is 7.5 km (Wingham et al., 2006). The SIRAL altimeter (Ku-band) on board CryoSat-2 combines a pulse-limited radar altimeter, with synthetic aperture and interferometric signal processing (320 MHz). The footprint of CS2 is pulse-Doppler-limited ∼300 m along track and pulse-limited ∼1500 m across the track of the beam, with measurements spaced at ∼300 m intervals (Wingham et al., 2006). CryoSat-2 is still active.

CryoSat-2 sea ice freeboard data for March 2010-2020 were obtained from the Lognormal Altimeter Retracker Model (LARM) that accounts for variable roughness as described in (Landy et al., 2020). The sea ice freeboards obtained from this algorithm have been demonstrated to perform well over thin sea ice and are therefore appropriate for a predominantly seasonal ice regime such as Baffin Bay (Landy et al., 2020).

CryoSat-2 radar freeboard data are also available from the ESA CCI record and we have obtained radar freeboard data on the satellite swath for March 2011-2017 from https://climate.esa.int/en/odp/#/dashboard (Hendricks et al., 2018a). This dataset shows some differences to the LARM processing, including significant variability between the two products along the coast. However, it is included to examine the impact of different CryoSat-2 freeboard products on derived ice thickness and to provide a direct comparison with the earlier Envisat CCI data. The CryoSat-2 CCI product includes a freeboard uncertainty estimate.

### 2.1.4 ICESat-2

ICESat-2 (IS2) was launched in September 2018 by NASA. The satellite orbits up to a latitudinal limit of 88° on a 91-day repeat period with subcycles of 29 and 33 days. ICESat-2 carries the Advanced Topographic Laser Altimeter System (ATLAS) laser altimeter that operates in a split-beam configuration of three beam pairs which each include a strong and a weak beam (all at a wavelength of 532 nm). Each beam pair is separated by ∼3 km across-track, with a pair spacing of 90 m (Markus

et al., 2017). The multiple beam pairs and high along-track sampling rate (10 kHz shots every 70 cm) provide improved spatial coverage over existing satellite altimetry missions over sea ice. The small footprint diameter of each of the beams (~11 m) is useful for sea surface height measurements in the often narrow leads needed for sea ice thickness retrievals (Kwok et al., 2019). The along-track freeboard height is computed by differencing sea ice heights from individual segments (produced through an aggregation of 150 photons along each of the beams) and local reference sea surface height determined from the lead height

estimates within each beam (Kwok et al., 2020).

ICESat-2 L3A ATL10 total freeboard data (release 003) were retrieved from the NSIDC (https://nsidc.org/data/icesat-2/data-sets) for March 2019 and 2020 (Kwok et al., 2020). The along-track freeboard estimates from all six IS2 beams was applied in this study. The product includes a freeboard uncertainty estimate.

## 2.2 Snow data and redistribution

### 2.2.1 Snow depth and density

We apply three snow depth products on the satellite altimetry data: the Warren climatology (W99), a passive microwave (PMW) snow depth product and modelled snow depth forced by reanalysis snowfall. The Warren climatology (Warren et al., 1999, W99) (Fig. 3a) is based on field measurements collected in the central Arctic Ocean. Webster et al. (2014) showed that the climatology is not representative of the Arctic-wide snow cover characteristics of recent years when compared against

contemporary measurements from NASA's Operation IceBridge (OIB) airborne surveys. Analysis of the OIB snow depth estimates suggested that snow depth on first year ice is approximately 50% the snow depth shown by W99 in those same regions (Kurtz and Farrell, 2011). As the climatology does not cover Baffin Bay, the regionally adapted W99 snow depth estimate for Baffin Bay is taken as the mean of snow depth on Pan-Arctic FYI in the W99 climatology and applied as a constant value over space and time. We multiply the mean FYI W99 snow depth climatology by 0.5 to account for the thinning

of the FYI snow pack following Laxon et al. (2013). A constant snow depth value is unrealistic, but it acts as a reference for other snow products and to compare with previous studies in which W99 has been used as the default snow depth dataset.

The PMW snow depth product was created from DMSP SSM/I-SSMIS brightness temperature measurements. DMSP brightness temperatures were chosen over AMSR brightness temperate data, because DMSP provides a continuous record over the study period and the PMW snow depth from DMSP better reconciles sea ice thickness from CryoSat-2 and ICESat-2

over the coinciding period of these two sensors (2019-2020). Moreover, it has been shown that DMSP provides a similar snow depth distribution and there are no anomalous biases compared to AMSR-E in Baffin Bay (Landy et al., 2017). To create the PMW snow depth product over seasonal ice regions (Fig. 3b), the DMSP SSM/I-SSMIS Pathfinder Daily EASE-Grid (25km x 25km) Brightness Temperatures V2 at 37 GHz and 19 GHz vertical polarization were retrieved from NSIDC (https://nsidc.org/data/NSIDC-0032/versions/2). Daily PMW snow depth (2003-2020) was retrieved from the brightness tem-

peratures according to Markus and Cavalieri (1998). This method relies on a strong correlation between decreasing brightness temperature with increasing snow depth from increased volume scattering in the snow pack (Rango et al., 1979). Scattering

decreases with increasing wavelength; therefore the ratio between brightness temperatures at 37 GHz and 19 GHz is used to determine snow depth (Markus and Cavalieri, 1998).

Modelled snow depths used in this study come from the recently developed Lagrangian snow-evolution model SnowModel-LG (Liston et al., 2020; Stroeve et al., 2020) (Fig. 3c). The model output used here were forced by MERRA2 atmospheric reanalyses and NSIDC sea ice parcel concentration and trajectory datasets to produce daily snow distributions on a 25-km x 25-km grid (Liston et al., 2020). Processes including snow blowing, snow density evolution and ice dynamics are accounted for (Liston et al., 2020). We also utilized snow depth estimates from the NASA Eulerian Snow On Sea Ice Model (NESOSIM) (Petty et al., 2018), which uses similar large-scale reanalysis and satellite data input to SnowModel-LG, but more simple parameterizations of snow accumulation and loss. NESOSIM snow depth estimates were applied as a second check on the effect of dynamic snow model results on sea ice thickness results in Baffin Bay.

A fifth snow depth product was created from CryoSat-2 and ICESat-2 freeboard observations, following the method of (Kwok et al., 2020) (Fig. 3d). Snow depth is expressed as the difference between the total freeboard (measured by ICESat-2) and sea ice freeboard (the radar freeboard measured by CryoSat-2 plus a correction for the propagation speed through the snowpack). This method can only be applied for the overlapping period between the two satellites (2019-2020) and is thus not used for sea ice thickness processing but used as one of multiple references for evaluating the other snow and derived sea ice thickness products.

To calculate the sea ice thickness from the freeboard data, also the snow density is needed. Two snow density products were used. Firstly, a spatially constant snow density for each given day of the year was retrieved from the density observations from Warren et al. (1999). Secondly, the daily snow density outputs from SnowModel-LG were applied.

### 2.2.2 Snow redistribution

As the horizontal resolution of the snow depth products ($\sim$25-100 km) is much lower than the resolution of the along-track freeboard measurements ($\sim$15m - 2km), snow redistribution provides a simple, albeit rudimentary, attempt to represent small-scale variability not captured in the large-scale snow depth products.

In this study we apply and test the effect of two snow redistribution methods together with a non-redistribution of snow. The first redistribution function is the sigmoidal function by Kwok and Cunningham (2008). This function determines the effective snow depth as a fraction of the total freeboard. First, one determines the total freeboard-snow depth ratio. If this ratio is above 1.3, the effective snow depth is the same as the large-scale snow depth. Below 1.3, the effective snow depth-snow depth ratio is determined from the sigmoidal function based on the freeboard-snow depth ratio (Kwok and Cunningham, 2008). The functional fit was based on heuristic ideas regarding snow accumulation/loss (thinner ice is generally younger so has had less time for snow to accumulate on) and is not based on empirical analysis (high-resolution snow/freeboard data was generally lacking when it was first developed). This function generally removes snow and does not attempt to conserve snow depth (i.e., mass) so can be considered as an aggregate snow loss-term on the large-scale snow depth.

The second redistribution we applied is the piecewise function that was determined based on airborne measurements of total freeboard from the Airborne Topographic Mapper (ATM) and snow depth from Snow Radar from NASA's Operation IceBridge

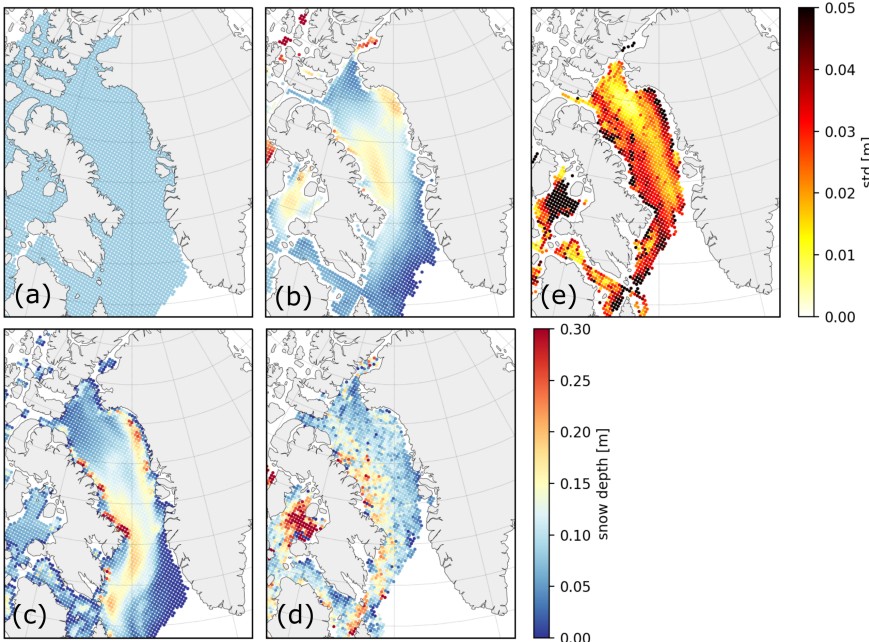

**Figure 3.** March mean of snow depth products. a) W99 climatology (constant), b) PMW snow depth product (2001-2020), c) SnowModel-LG (2001-2020), d) CryoSat-2/ICESat-2 product (2019-2020) and the standard deviation between the different products (e).

airborne campaigns (Petty et al., 2020). This function was first adopted by Kurtz et al. (2009) to represent the full snow depth distribution on the regional scale, based on empirical evidence of a linear relationship between freeboard and snow depth up to a certain (threshold) freeboard. This redistribution function is applied iteratively to conserve snow depth within the given large-scale grid cell. This piecewise function was shown to improve the representation of sub grid-scale variability over no
redistribution or other functional fits in the OIB data (Petty et al., 2020).

The sigmoidal function (Kwok and Cunningham, 2008) and the piecewise function (Petty et al., 2020) were designed to redistribute snow depth for high-resolution laser altimetry data. However, snow depth redistribution schemes have not yet been tested for radar altimetry data. We assessed whether there is a requirement and any benefit in using a radar freeboard snow redistribution scheme. We follow the methods outlined in Kurtz et al. (2009) to evaluate this, which is further described in
Supplementary Materials S1.

The radar freeboard estimates did not show a relation with the snow depth on the length-scale of the pulse-limited footprint of the Envisat or the SAR-focused footprint of CryoSat-2 radar altimeters (Fig. S1). This shows that snow redistribution on radar altimetry freeboard data would not improve the conversion from ice freeboard to thickness, so we did not apply a redistribution function to the snow depth. For Envisat and CryoSat-2 data we simply used the snow depths at their native resolution (25 km)
to convert freeboards to ice thicknesses.

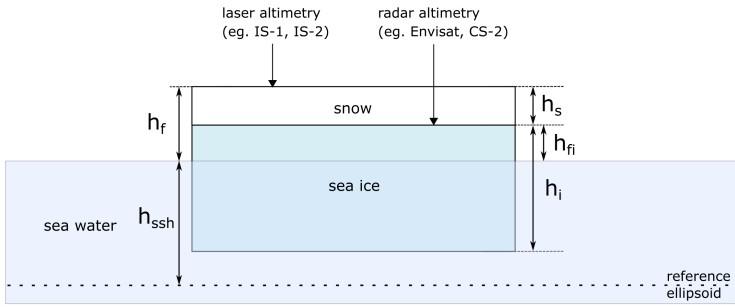

**Figure 4.** Schematic of sea ice floe, floating in hydrostatic equilibrium, and discussed variables.

## 2.3 Sea ice thickness

Along-track freeboard samples from all satellite altimetry products were discarded where the distance to land was less than 10 km or where the distance to the closest lead was more than 200 km to avoid land contamination and uncertain sea surface height from interpolation between leads.

### 235 2.3.1 Laser altimetry

When the snow is not redistributed on laser altimetry data, the large-scale snow depth can be larger than the small-scale total freeboard at individual along-track footprints, resulting in negative sea ice freeboards. Sea ice freeboard can be negative when the snowpack provides such a load that it depresses the ice surface below sea level. However, when the snow depth $h_s$ exceeds the ratio $\frac{\rho_w}{(\rho_w - \rho_s)} h_{f_{total}}$ - with $\rho_w$ density of sea water (1024 kg m$^{-3}$), $\rho_s$ the density of snow, and $h_{f_{total}}$ the total sea ice plus 240 snow freeboard – this results in a physically impossible negative sea ice thickness. We assume that the snow depth measurement for these samples is invalid and decrease the snow depth to be equal to the total freeboard measurement. This is true for 17-25% of the ICESat measurements and 8-9% of the ICESat-2 measurements when no redistribution was applied.

We calculated sea ice thickness (Fig. 4) from the along-track laser freeboard data with the W99 snow depth, the PMW snow depth and the SnowModel-LG snow depth, with the W99 and SnowModel-LG snow density products, and without 245 redistribution of snow, with the sigmoidal function of effective snow depth (Kwok and Cunningham, 2008) and with the piecewise snow depth redistribution (Petty et al., 2020).

Sea ice thickness $h_i$ was estimated from ICESat and ICESat-2 along-track freeboard observations using:

$$h_i = (\frac{\rho_w}{\rho_w - \rho_i}) h_{f_{total}} - (\frac{\rho_w - \rho_s}{\rho_w - \rho_i}) h_s \tag{1}$$

where $h_{f_{total}}$ is the total sea ice plus snow freeboard, $h_s$ is the depth of the snow layer, and $\rho_w$, $\rho_i$, and $\rho_s$ are the densities of 250 sea water, sea ice, and snow respectively. Sea water density was taken as 1024 kg m$^{-3}$ and sea ice density was obtained from an ice thickness-dependent parameterization: $\rho_i(h_i) = 936 - 18h_i^{0.5}$ kg m$^{-3}$ (Kovacs, 1996).

### 2.3.2 Radar altimetry

Sea ice thickness was estimated from Envisat and CryoSat-2 along-track radar freeboard observations with the W99 snow depth, the PMW snow depth product and the SnowModel-LG snow depth, with the W99 and SnowModel-LG snow density products, and without redistribution of snow using:

$$h_i = (\frac{\rho_w}{\rho_w - \rho_i}) h_{f_i} + (\frac{\rho_s}{\rho_w - \rho_i}) h_s \qquad (2)$$

where $h_{f_i}$ is the ice-only freeboard, because the radar is assumed to penetrate the snow pack. To account for the lower propagation speed of the radar wave through the snowpack, a correction is applied to the radar freeboard:

$$h_{f_i} = h_{f_{radar}} + h_s(\frac{c}{c_s} - 1) \qquad (3)$$

where $c_s$ is the speed of light through snow, parameterized by $c_s = c(1 + 5.1 \cdot 10^{-4} \rho_s)^{-1.5}$ (Ulaby et al., 1986). Sea water density ($\rho_w$) was taken as 1024 kg m$^{-3}$ and sea ice density was obtained from an ice thickness-dependent parameterization: $\rho_i(h_i) = 936 - 18h_i^{0.5}$ kg m$^{-3}$ (Kovacs, 1996).

### 2.3.3 Uncertainty and gridding

The along-track sea ice thickness values were gridded on a 12x12 km (CryoSat-2 and ICESat-2) and 25x25 km (ICESat and Envisat, which provide a lower number of observations) EASE-grid using a mean filter inverse-linearly weighted by the sample uncertainty and distance (Geiger et al., 2015) up to a distance of twice the resolution of the grid.

The sea ice thickness sample uncertainty was estimated by accounting for individual uncertainties in freeboard, snow depth and snow, sea ice and ocean water density (Landy et al., 2017). The individual uncertainty estimates for snow depth, snow density, sea ice density, and ocean water density from Landy et al. (2017) were applied. The random individual uncertainties in sea ice density and ocean water density and speckle noise (for the radar data) were assumed to be uncorrelated and could be scaled by the number of observations within the grid cell. The individual uncertainties in the snow depth and snow density products were assumed to be correlated within the grid cell, as they were all interpolated from gridded products. The freeboard uncertainty for each of the sensors was assumed to be correlated within a single track, but the separate tracks are uncorrelated, so the uncertainty was scaled by the number of tracks. The individual uncertainties were combined, using Gaussian propagation of uncertainty, to generate a single uncertainty estimate for each sea ice thickness grid cell. The uncertainty in the mean climatological sea ice thickness was found by taking the spatial average of the uncertainty estimates.

### 2.4 Canadian Ice Service Charts

The Canadian Ice Service (CIS) Charts provide a continuous estimate of sea ice concentration, stage of development (e.g. new ice, first-year ice (FYI), multi-year ice (MYI)) and forms of ice in the Canadian Arctic from 1968 to present. The ice charts are available weekly in summer (June-November) since the start of data collection) and monthly (1980-2005), bi-weekly (2006-2011) or weekly (2012-present) in winter (December-May). During the first decades the data collections consisted mainly of

**Table 1.** CIS sea ice stages of development and their thickness ranges, with between brackets the used average thickness of each range. *Range given by NSIDC definition.

| | |
|---|---|
| 1. New ice | <10 cm (5 cm) |
| 2. Nilas | <10 cm (5 cm) |
| 4. Grey ice | 10-15 cm (12.5 cm) |
| 5. Grey white ice | 15-30 cm (22.5 cm) |
| 7. Thin first year ice | 30-70 cm (50 cm) |
| 10. Medium FYI | 70-120 cm (95 cm) |
| 40. Thick FYI | >120 cm (120 cm) |
| 70. Old ice | 200-400 cm (300 cm)* |
| 80. Second year ice | 200-400 cm (300 cm)* |

ship and airborne observations. Over time the CIS integrated several data sources in the ice charts including satellite remote sensing data. The largest improvement in accuracy was achieved in 1996 with the inclusion of near-real-time satellite data from the Canadian remote sensing Earth observation satellite program RADARSAT.

We make an estimate of sea ice thickness from the stage of development estimates from the CIS charts. The CIS charts were retrieved from https://iceweb1.cis.ec.gc.ca/Archive/page1.xhtml?lang=en. We convert CIS .e00 vector files into 5 km resolution gridded 'raster' datasets. The charts provide the five most present stages of development and their partial concentration within each polygon. The stages of development are given with a range of thicknesses and the averages of these ranges are taken (Table 1). To estimate sea ice thickness from the ice charts, the mean of the thickness for each stage is taken weighted by the

partial concentration of the stages of development. The uncertainty is estimated by the range of thicknesses given (Table 1). The resulting product does not provide the actual sea ice thickness and the absolute values may not be reliable, but the product can be used to investigate relative spatial and temporal patterns in estimated sea ice thickness. Here we only use ice chart data following the launch of RADARSAT in 1996, so that results are consistent across the analysed time period.

## 3 Results

### 3.1 Mean sea ice thickness and trends

As there are few in-situ observations of snow depth or sea ice thickness available within Baffin Bay, it is not possible to determine unambiguously which of the snow depth products and redistribution methods give the most accurate estimation of sea ice thickness from satellite altimetry. We therefore look at the mean of the processing methods to illustrate the long-term inter-mission sea ice thickness in Baffin Bay (Fig. 5).

The March mean sea ice thickness of all the satellite records (Fig. 5a-d) confirms a strong west-east asymmetry in sea ice thickness across the bay (Landy et al., 2017). However, the magnitude of sea ice thickness is quite different between the

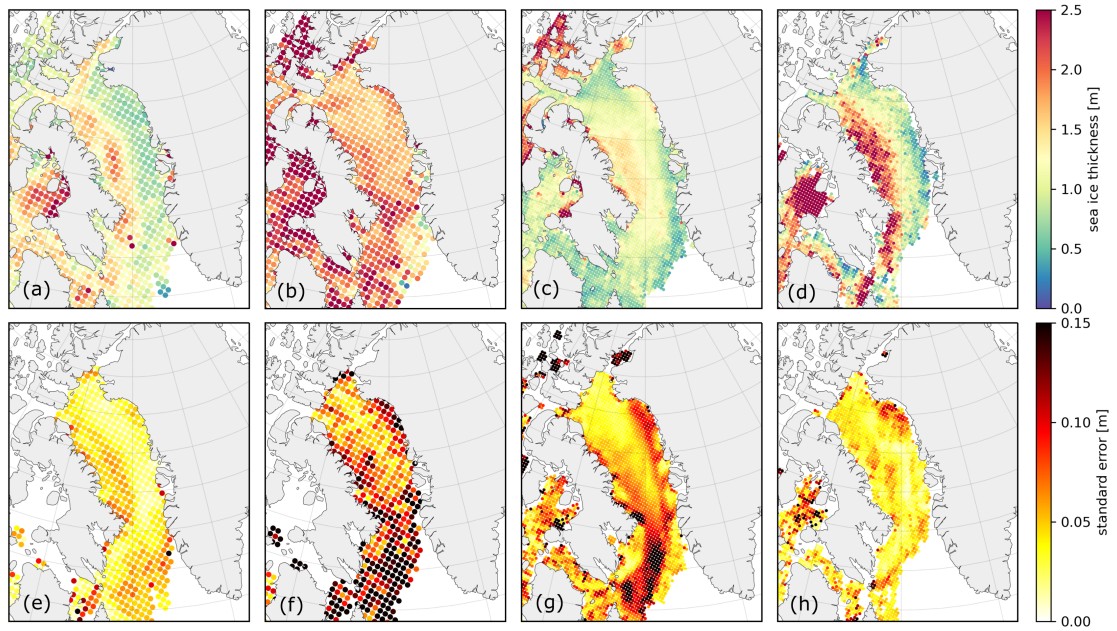

**Figure 5.** March mean sea ice thickness, mean and standard error of mean of all processing methods for a,e) ICESat (2003-2009), b,f) Envisat (2003-2012), c,g) CryoSat-2 LARM (2011-2020), and d,h) ICESat-2 (2019-2020).

different satellite altimetry products. The March mean sea ice thickness of the satellite products cover different periods of time in the record but overlapping time periods of satellites also give different magnitudes of sea ice thickness. The mean March sea ice thickness from Envisat (1.86 ± 0.44 m) is much higher than for the other satellite products (1.12 ± 0.47 m, 1.13 ± 0.47 m and 1.30 ± 0.42 m for ICESat, CryoSat-2 LARM and ICESat-2 respectively). The derived ice thickness in the west of the bay from ICESat-2 is a lot thicker than for the other satellite altimetry records. This results in a much stronger asymmetry. The CryoSat-2 sea ice thickness record shows a thicker region of sea ice in Melville Bay, in the northeast of Baffin Bay, which is typically not visible in the other satellite altimetry records. This thicker region is present in the CryoSat-2 sea ice thickness results for March 2019 but is not evident in the ICEsat-2 sea ice thickness results for the same year.

There is no significant trend in the mean sea ice thickness (mean of all processing methods) in Baffin Bay from ICESat and CryoSat-2 LARM (2003-2020).

### 3.2 Spread in results from processing methods

March mean ICESat sea ice thickness products from the eighteen different processing techniques show a similar spatial pattern of thick sea ice in the west of Baffin Bay and thinner sea ice in the east (Fig. 5a). However, the magnitude of ice thickness depends on the chosen snow depth product and redistribution method (Table 2). Generally, the PMW product results in thinner sea ice than the constant W99 snow depth value applied for FYI (16% with no redistribution, 9% with the sigmoidal function,

and 18% with the piecewise function). SnowModel-LG product results in slightly thinner sea ice thickness than the W99 product (9% with no redistribution, 5% with the sigmoidal function, and 10% with the piecewise function). Redistributing the snow depth results in a thinner sea ice pack with the piecewise function and a thicker sea ice pack with the sigmoidal function for all snow depth products. ICESat March mean sea ice thickness using the PMW snow depth product with the sigmoidal and the piecewise redistribution gives respectively 12% thicker and 2% thinner sea ice than without the redistribution. The SnowModel-LG snow density product results in on average 4.6% thinner sea ice than the W99 snow density product.

The chosen snow depth product has a significant effect on the spatial pattern of the CryoSat-2 March sea ice thickness (see Supplementary Materials S2). SnowModel-LG snow depth results in a different spatial pattern in ice thickness than the other two products, with thicker ice along the coast and near the sea ice margin. This is because the SnowModel-LG depths are larger at the ice edge meaning the ice freeboards are more depressed by the snow load and final estimated thicknesses are higher. The mean sea ice thickness with the SnowModel-LG snow depth (1.24 m) is almost equal as with the PMW snow depth (1.23 m). The SnowModel-LG snow density product results in on average 10% thinner sea ice than the W99 snow density product.

The choice of CryoSat-2 radar freeboard product has a significant effect on the magnitude of sea ice thickness. Sea ice thickness is ~30% thicker when generated with the CCI freeboard product than with the LARM freeboard product (see supplementary material).

The time series of mean March sea ice thickness within Baffin Bay is shown in Fig. 6 for all four sensors and the different processing methods shown as envelopes. The magnitude and the sign of the trend in mean March sea ice thickness from ICESat and CryoSat-2 LARM (2003-2020) depends on the snow depth product and redistribution method used (Table 3). The sea ice thickness trend ranges from a thinning of -23 cm/dec to a thickening 17 cm/dec. Only the sea ice thickness with the constant and not dynamically varying W99 snow depth results in significant trends (p<0.05). The trend is not significant for any of the other snow depth products. The trend in March sea ice thickness from ICESat and CryoSat-2 CCI (2003-2017) is higher and ranges from -4.9 cm/dec (W99 snow depth, Sigmoidal, SnowModel-LG density) to 36.1 cm/dec (PMW snow depth, no redistribution, W99 density). The March sea ice thickness trend from Envisat CCI and CryoSat-2 CCI (2003-2017) ranges from -68 cm/dec to -38.6 cm/dec.

Moreover, the spatial pattern of sea ice thickness trends vary for different combinations of snow depth and density products and snow redistribution methods (Fig. 7). The W99 snow depth and density with sigmoidal redistribution SIT (Fig. 7a) shows a strong asymmetrical thinning pattern, with strong thinning in the west of Baffin Bay and no thinning in the east. The PMW snow depth with piecewise redistribution and W99 snow density (Fig. 7b) shows thickening in almost all of Baffin Bay, with some thinning in the North Water Polynya region. The SnowModel-LG snow depth and density with piecewise redistribution (Fig. 7c) shows thinning in the northwest of the bay and thickening in the east and south.

### 3.3 Asymmetry and asymmetry trends

The sea ice thickness asymmetry across Baffin Bay is determined from the difference in mean sea ice thickness in western Baffin Bay and eastern Baffin Bay (Fig. 1). Although there is apparent west-east asymmetry in sea ice thickness across Baffin Bay in all products, the level of west-east sea ice thickness asymmetry across Baffin Bay depends on snow depth product

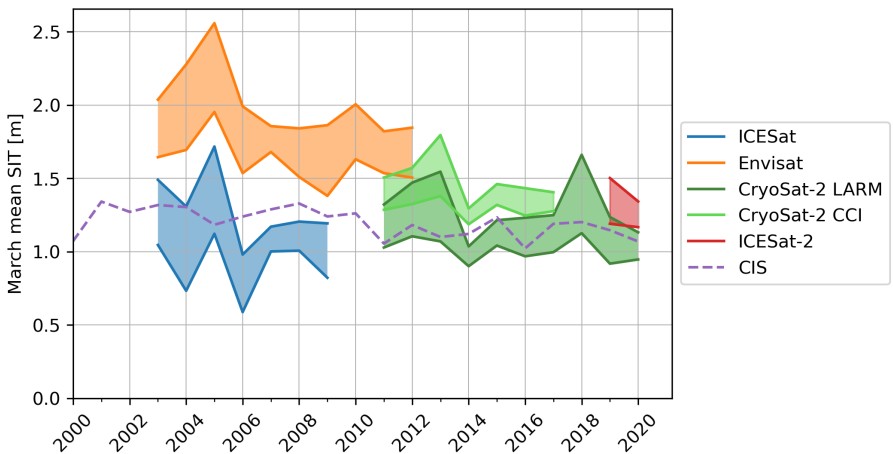

**Figure 6.** Timeseries of March mean sea ice thickness from the different satellite altimeters and the estimated sea ice thickness from CIS ice charts. The envelopes show the range of outcomes from different processing methods.

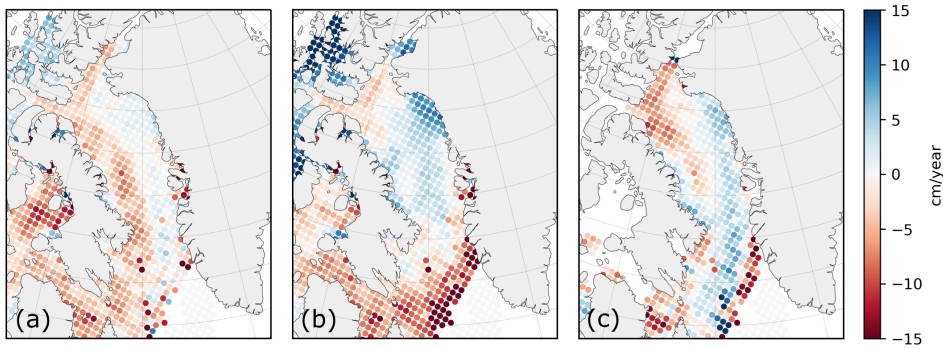

**Figure 7.** March sea ice thickness trends (2003-2020, ICESat + CryoSat-2 LARM) with different processing methods. a) W99 snow depth + sigmoidal redistribution + W99 snow density, b) PMW snow depth + piecewise redistribution + W99 snow density, c) SnowModel-LG snow depth + piecewise redistribution + SnowModel-LG snow density.

**Table 2.** ICESat mean sea ice thickness (2003-2009) for different snow depth products and snow redistribution methods. The first figure is with the W99 snow density value and the italic figure is with the SnowModel-LG snow density product.

|  | W99 | PMW | SnowModel-LG |
|---|---|---|---|
| No redistribution | 1.23 ± 0.45 m | 1.03 ± 0.45 m | 1.12 ± 0.44 m |
|  | *1.18 ± 0.49 m* | *0.98 ± 0.49 m* | *1.07 ± 0.49 m* |
| Sigmoidal function | 1.28 ± 0.44 m | 1.17 ± 0.45 m | 1.22 ± 0.44 m |
|  | *1.23 ± 0.49 m* | *1.10 ± 0.49 m* | *1.17 ± 0.49 m* |
| Piecewise function | 1.24 ± 0.45 m | 1.02 ± 0.45 m | 1.12 ± 0.45 m |
|  | *1.19 ± 0.49 m* | *0.94 ± 0.49 m* | *1.07 ± 0.49 m* |

**Table 3.** Trend in mean March sea ice thickness for ICESat and CryoSat-2 LARM (2003-2020) for different snow depth products and snow redistribution methods. The first figure is with the W99 snow density value and the italic figure is with the SnowModel-LG snow density product. The trend varies from -22.5 cm/dec to 17.0 cm/dec across the full range of processing methods (results in bold are significant within p<0.05).

|  | W99 | PMW | SnowModel-LG |
|---|---|---|---|
| No redistribution | -12.9 cm/dec | 15.6 cm/dec | 4.2 cm/dec |
|  | ***-19.0 cm/dec*** | *8.3 cm/dec* | *-1.5 cm/dec* |
| Sigmoidal function | **-16.5 cm/dec** | 4.8 cm/dec | -4.0 cm/dec |
|  | ***-22.5 cm/dec*** | *-2.5 cm/dec* | *-9.6 cm/dec* |
| Piecewise function | -14.0 cm/dec | 17.0 cm/dec | 3.8 cm/dec |
|  | ***-19.5 cm/dec*** | *11.2 cm/dec* | *-1.3 cm/dec* |

and redistribution technique (Fig. 8). There is also yearly variability in the strength of asymmetry. Trends in asymmetry, determined from the combined ICESat and CryoSat-2 LARM sea ice thickness data record using different snow depth products and redistribution methods, are summarized in Table 4. The mean of all the processing methods gives a significant decrease in west-east asymmetry over time of -28.3 cm/dec, caused by a thinning in the west of -17.6 cm/dec and a thickening in the east of 10.8 cm/dec. The trend in asymmetry depends on which snow data has been used. When the snow depth is not redistributed, both the PMW and the SnowModel-LG snow depth products with W99 snow density result in a significant decrease in sea ice thickness asymmetry across the bay over the period 2003-2020. The SnowModel-LG product also results in a significant decrease in sea ice thickness asymmetry over time with redistribution of snow depth. However, the PMW product with the sigmoidal and with the piecewise snow depth redistribution function do not result in a significant reduction in asymmetry. This shows that any trends in asymmetry are dependent on the chosen snow depth and density product and redistribution method. The asymmetry trends when combining ICESat and CryoSat-2 CCI sea ice thickness results are slightly stronger and range from -9 cm/dec (PMW, Piecewise) to -42 cm/dec (SnowModel-LG, no redistribution).

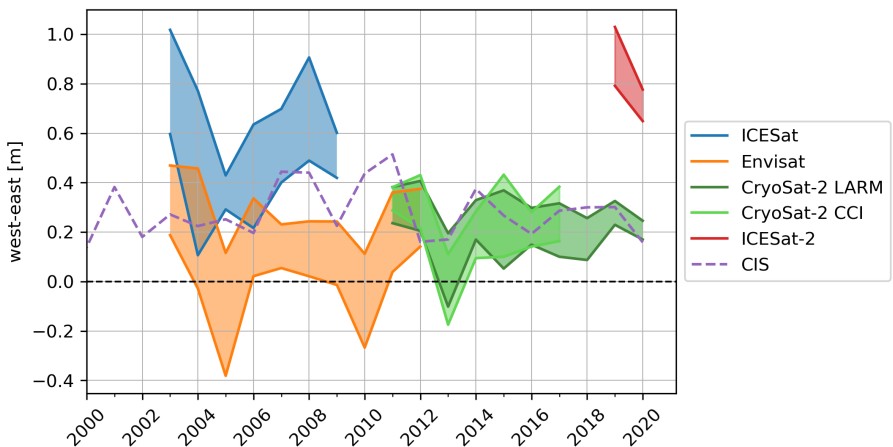

**Figure 8.** Timeseries of west-east sea ice thickness asymmetry from the different satellite altimeters and the estimated sea ice thickness from CIS ice charts. The envelopes show the range of outcomes from different processing methods.

**Table 4.** Trend in west-east asymmetry for ICESat and CryoSat-2 LARM (2003-2020) for different snow depth products and snow redistribution methods. The first figure is with the W99 snow density value and the italic figure is with the SnowModel-LG snow density product. The trends in asymmetry range from –41.6 cm/decade to –9.4 cm/decade (results in bold are significant within p<0.05).

|  | W99 | PMW | SnowModel-LG |
|---|---|---|---|
| No redistribution | **-32.4 cm/dec** | **-14.7 cm/dec** | **-41.6 cm/dec** |
|  | *-33.3 cm/dec* | *-12.9 cm/dec* | **-39.5 cm/dec** |
| Sigmoidal function | **-31.2 cm/dec** | **-15.3 cm/dec** | **-40.0 cm/dec** |
|  | *-32.0 cm/dec* | *-13.5 cm/dec* | **-37.9 cm/dec** |
| Piecewise function | **-34.4 cm/dec** | -11.9 cm/dec | **-38.3 cm/dec** |
|  | *-35.4 cm/dec* | *-9.4 cm/dec* | **-36.3 cm/dec** |

## 3.4 Canadian Ice Service Chart record

The estimated thickness and spatial patterns from the CIS ice charts are consistent with the ICESat and CryoSat-2 sea ice thickness results (Fig. 9a). The mean thicknesses of ICESat and the CIS estimated thicknesses are $1.12 \pm 0.47$ m and $1.26 \pm 0.26$ m respectively, for the coinciding period of 2003-2009. The mean thicknesses of CryoSat-2 LARM and the CIS thicknesses are $1.13 \pm 0.47$ m and $1.13 \pm 0.26$ m respectively, for the coinciding period of 2011-2020.

Trends in FYI and MYI can be determined by the change in partial concentration of these ice types from the CIS charts (Fig. 10). There has been a negative trend in MYI in the western part of the bay from ∼37% to ∼9% between 1996 and 2020, with this older ice being replaced by increasing FYI in most of Baffin Bay. Near the west coast, there has been some increase in MYI and decrease in FYI. In the northern part of the bay, there has been a decrease in both FYI and MYI, with these mature

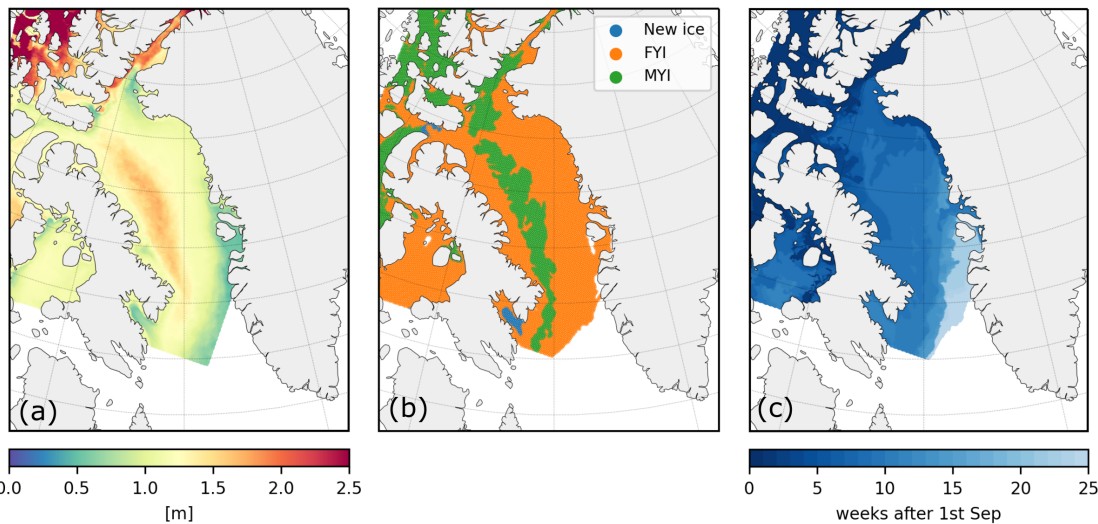

**Figure 9.** CIS ice chart (a) mean estimated thickness (1996-2020), (b) oldest stage of development 18-03-2019, and (c) freezeup start date 2018-2019.

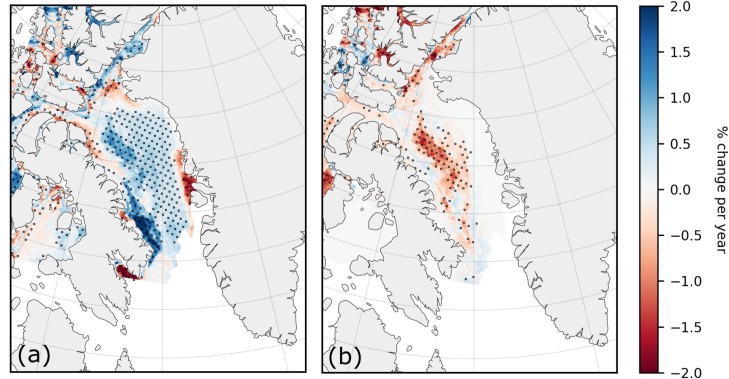

**Figure 10.** Trend in first-year ice (a) and multi-year ice (b) from the Canadian Ice Service charts (1996-2020). Dotted areas show significant trends within p<0.05.

ice types being replaced by new and young first-year sea ice. The replacement of MYI by FYI since the 2000s suggests Baffin Bay has undergone a transition where less MYI is being imported through Nares Strait and/or the Canadian Arctic Archipelago and more FYI is grown in place. The reduction in both ice types in northern Baffin Bay suggests an increase in the size and production of new/young sea ice types in the North Water Polynya region. These trends in ice type have contributed to a general thinning of sea ice in Baffin Bay since 1996, according to the ice chart stage-of-development, of 7 cm per decade (p<0.05).

## 4 Discussion

### 4.1 Mean state and trends

The March mean sea ice thickness of all the satellite records showed a strong west-east asymmetry in sea ice thickness across the bay. The characteristic west-east asymmetry in sea ice thickness that was found within Baffin Bay – both in the altimetry thickness fields and the sea ice charts – can be explained by a combination of stage of development of the ice and freezeup date. In spring the main stage of development within the bay is first-year ice, with a band of multi-year ice generally present along the west coast (Fig. 9b). As the bay is ice free in summer, this ice must be imported through Nares Strait and the Canadian Arctic Archipelago channels into Baffin Bay and drift with the Baffin Island Current along the west coast of the Bay (Bi et al., 2019; Kwok, 2005). Secondly, the freezeup date in the western part of the bay is earlier in the season than in the eastern part of the bay (Fig. 9c), resulting in a longer period over which the ice can grow and thicken and more snow to accumulate. The sea ice generally drifts southwards along the western side of the bay (Tang et al., 2004), following the Baffin Island Current, and does not cross the bay to the east.

The CIS charts show a replacement of MYI with FYI within the bay in the first decades of the 21st century, with a decrease of MYI in the northwest of $\sim$37% to $\sim$9% between 1996 and 2020. This might be associated with a thinning of spring sea ice within Baffin Bay, but no significant change in sea ice thickness was found from the altimetry products within the error margins for all inter-mission processing methods for the limited period multi-mission altimetry data are available. It is possible the MYI entering and transiting Baffin Bay has always included decayed and thinner ice floes that are not appreciably thicker than FYI floes measured with altimetry.

The freezeup onset displays a trend with earlier freezeup in the north-east and later freezeup in the south-east of Baffin Bay over the past two decades. The freezeup date in the west of Baffin Bay does not display any obvious trend.

The stage of development from the CIS ice charts show strong agreement with the satellite altimetry sea ice patterns and offers the potential to be used as a proxy for the mean and regional distribution in sea ice thickness and spatial pattern (see Supplementary Materials S3). Despite this, the CIS ice charts show lower interannual variability compared to the satellite altimetry sea ice thickness products.

### 4.2 Intermission differences

The long-term record includes results from four separate satellite altimetry missions. The different satellite altimetry products give different results for sea ice thickness in Baffin Bay, even over the same period. The Envisat sea ice thickness for 2003-2009 (1.89 $\pm$ 0.44 m) is significantly higher than the ICESat sea ice thickness product (1.12 $\pm$ 0.47 m) over the same period. The Envisat sea ice thickness product for 2011-2012 (1.66 $\pm$ 0.43 m) is also significantly higher than the CS2 sea ice thickness product for the same period (1.19 $\pm$ 0.46 m). ICESat, CryoSat-2, ICESat-2 and the Canadian Ice Service charts over their entire period (1.12 $\pm$ 0.47 m, 1.13 $\pm$ 0.47 m, 1.30 $\pm$ 0.42 m, 1.20 $\pm$ 0.26 m respectively) all give similar lower mean sea ice thicknesses than Envisat (1.86 $\pm$ 0.44 m). Envisat shows a general tendency to overestimate sea ice thickness in FYI dominated regions (Kern et al., 2018), which could be due to a sampling bias of wider, thicker ice floes (Tilling et al., 2019). Therefore,

Envisat results might be overestimating sea ice thickness within Baffin Bay. The CCI Envisat freeboard is calibrated with focus on the central Arctic and the results suggest the CCI Envisat freeboard does not do as well in marginal regions such as Baffin Bay. This suggests further processing work on historic radar altimetry record may be required to create reliable sea ice thickness products in the seasonal ice zone or to evaluate thickness trends in regions that have transitioned from MYI- to FYI-dominated over the past few decades.

The similarities between missions are improving for the more recent satellite altimeters with a better agreement between CryoSat-2 (both LARM and CCI) and ICESat-2 (mean difference in sea ice thickness for overlapping years of 0.25 m, 24%) than Envisat and ICESat (mean difference in sea ice thickness for overlapping years of 0.77 m, 69%).

The CryoSat-2 sea ice thickness product shows a thick region in Melville Bay to the northeast of Baffin Bay in March for 7 of the 10 covered years (e.g. Fig. 8c), which is not as present in the sea ice thickness products from the other missions. A discussion of this feature is presented in the Supplementary Materials (S4).

A comparison between the CryoSat-2 LARM sea ice thickness product and the Alfred Wegener Institute CS2/SMOS data fusion product (Ricker et al., 2017) shows general thicker sea ice in the LARM CryoSat-2 sea ice thickness product (Supplementary Materials S5). The spatial distribution of sea ice thickness is similar. Comparisons of CryoSat-2 mean sea ice thickness with a model-ensemble based estimation in Baffin Bay (Min et al., 2021) shows a similar spatial pattern and magnitude in sea ice thickness for most of Baffin Bay (Supplementary Materials S5). The higher sea ice thickness in the north east of Baffin Bay in the LARM CryoSat-2 sea ice thickness from this study is not present in the Min et al. (2021) sea ice thickness ensemble product.

### 4.3 Different processing methods

The long-term record of sea ice thickness from multiple satellite altimetry missions processed with different snow depth and density products and snow redistribution methods can lead to significantly different results, and therefore influence conclusions on total and regional sea ice thickness trends. The selected snow depth product influences both the found mean sea ice thickness, spatial patterns and trends. Processing decisions can introduce regional biases that, at least in thin ice areas, obscure decadal trends and patterns. Care must be taken to estimate true product uncertainty envelopes. For instance, across the wide range of processing options tested here (for ICESat and CryoSat-2) we find a mean uncertainty envelope around the March mean sea ice thickness of 13% and a range of possible multi-mission trends of -23 cm/decade to +17 cm/decade.

When comparing the snow depth products to a determination of snow depth from the difference between CryoSat-2 radar freeboard and ICESat-2 laser freeboard in March 2019 and 2020 (Kwok et al., 2020) (Fig. 3d), the PMW snow depth product shows the most similar pattern and magnitude. SnowModel-LG shows thicker snow depths along the coasts and the sea ice margin, and much thinner snow depths in the centre of the bay. SnowModel-LG also does not capture the west-east asymmetry that the PMW snow depth product shows and would be expected because of the longer period for snow to accumulate on the older ice in the west. Another snow model (NESOSIM, Petty et al. (2018)) was compared and shows a similar pattern in snow depth to SnowModel-LG but with thicker snow depth in the north of Baffin Bay (see S2). As there are no direct observations

of snow depth within Baffin Bay, a selection of the best snow depth product cannot be made. None of the snow depth products show a good reconciliation in sea ice thickness between the temporarily overlapping CryoSat-2 and ICESat-2 products.

A positive snow depth bias in the snow depth product has an opposite effect on sea ice thickness estimations from laser and radar altimetry. Sea ice thickness determined from laser altimetry would be underestimated, as a larger part of the total freeboard is sea ice than suggested by the snow depth product. Sea ice thickness determined from radar altimetry would be overestimated for two reasons. Firstly, the correction for radar wave propagation speed through the snow layer will be higher than it should be, overestimating the sea ice freeboard. Secondly, the assumed snow layer would weight down and suppress the

ice underwater more than the actual snow layer does, resulting in a larger thickness of sea ice extending below the water level and an overestimation of total sea ice thickness. This can also be understood from Equation 2: with greater assumed snow depth $h_s$, the first term would increase due to the snow propagation correction (Equation 3) and the second term would increase due to the adjustment of hydrostatic balance. In this study, we have determined sea ice thickness trends by combining results from the ICESat laser altimetry mission (2003-2009) and the CryoSat-2 radar altimetry mission (2011-2020). Applying a product

with a positive snow depth bias in a region would cause an underestimation of sea ice thickness in the ICESat period and an overestimation of sea ice thickness in the CryoSat-2 period, which would lead to an overestimation of (more positive) sea ice thickness trend in this region. Correspondingly, a negative snow depth bias causes an underestimation of (more negative) sea ice thickness trend.

   The trends in west-east sea ice thickness asymmetry strongly depend on the chosen processing methods and snow depth

product, for this thin seasonal sea ice region. Most of the processing methods exhibit a significant trend in west-east sea ice thickness asymmetry across the bay, agreeing with the finding of Landy et al. (2017). The only method that does not show a significant trend is the PMW snow depth with piecewise snow redistribution sea ice thickness.

## 4.4   Comparison with in-situ data

There are very few in-situ observations of snow depth or sea ice thickness available within Baffin Bay, which makes it chal-

lenging to determine which snow depth product and redistribution method is most accurate. However, since September 2004, a comprehensive observational program in Davis Strait has been providing sustained, long-term quantification of volume and freshwater transport between Baffin Bay and the Atlantic Ocean. As part of the program, Applied Physics Laboratory, University of Washington Mark 2 upward looking sonars (ULS) have been deployed to collect sea ice draft (the thickness of sea ice extending below the water level) measurements along Davis Strait (Curry et al., 2014; Lee et al., 2004). We compare dis-

tributions of draft observations at four locations along Davis Strait for March 2006-2008 with derived ice drafts from ICESat observations in the same period (Supplementary Materials S5) which show general agreement. The ICESat derived draft with the Warren '99 snow depth product, sigmoidal redistribution and the SnowModel-LG snow density product compares best with the mean of the ULS observed draft in Davis Strait for most years and locations (Supplementary Materials S5). However, there are only small differences between the Warren '99 assumed and PMW measured snow depths at the mooring locations, so ice

drafts estimated from these products are very similar.

The distributions of sea ice draft from the ULS observations and ICESat observations in the same period (Fig. 11) show larger sea ice drafts in the western part of Davis Strait than in the east. The ULS draft distribution captures two modes of sea ice, the very thin mode ($\sim 0.1$ m) which represents new sea ice (nilas) forming at the ice edge and a thicker mode ($\sim 1.1$ m for the west, $\sim 0.8$ m for the east) which represents level sea ice grown over a longer period and imported from Baffin Bay. When no redistribution of snow depth is applied (Fig. 11a,d), the first thin ice mode is captured well by ICESat. Applying a redistribution method (Fig. 11b,c,e,f) results in better capturing the second, thicker ice mode. This is caused by the fact that the redistribution methods redistribute snow by assigning a smaller snow depth to ICESat observations with lower freeboards and a larger snow depth to ICESat observations with larger freeboards. Therefore there will be a smaller snow depth on the very small freeboard observations when snow is redistributed, causing the sea ice thickness and draft to be larger for the thinner freeboard observations as opposed to when no redistribution is applied. This results in less observations of very thin sea ice drafts. The opposite is true for thicker freeboard observations, where redistribution results in more snow, so thinner sea ice thickness and draft when redistribution is applied.

Further discussion of the ULS draft observations with comparisons with ICESat sea ice draft with other snow depth products is available in Supplementary Materials S5. These are derived from only six local moorings and only one satellite mission close to the sea ice margin covering three years of the observation period, and cannot be viewed as a validation of our thickness product, or the snow depth products in Baffin Bay.

## 5 Conclusions

This study produced a long-term multi-mission record of spring Baffin Bay sea ice thickness (2003-2020) with multiple snow depth and density products. The record shows asymmetrical sea ice thickness for all satellite altimetry products, with thicker sea ice in the west and thinner sea ice in the east of the bay. There is no significant trend in mean sea ice thickness. However, this long-term record shows a significant decrease in the west-east asymmetry of -28.3 cm/dec.

This paper showed that there are inter-mission biases in sea ice thickness in this region. The Envisat sea ice thickness results ($1.86 \pm 0.44$ m) appear to overestimate the sea ice thickness compared to all other products ($1.12 \pm 0.47$ m, $1.13 \pm 0.47$ m, and $1.30 \pm 0.42$ m) in Baffin Bay. This suggests that the CCI Envisat freeboard may not be as effective over the thinner FYI of Baffin Bay as it is in the Central Arctic and suggests further processing work on historic radar altimetry data is needed to create reliable sea ice thickness products in the seasonal ice zone or in zones that have transitioned between multi- and first-year sea ice in recent decades.

Baffin Bay is part of the seasonal ice zone and it is becoming increasingly important to understand the trends and variability of sea ice in this region due to the rapid replacement of MYI by FYI that is both seen in Baffin Bay (a decrease of the concentration of MYI from 37% to 9% between 1996 and 2020) in this study from the CIS ice charts, and wider throughout the rest of the Arctic (Kwok, 2018; Mallett et al., 2020).

We have compared the effects of applying different snow depth products and snow redistribution methods on sea ice thickness calculations from satellite altimetry. We show that different data processing techniques in satellite altimetry can lead to

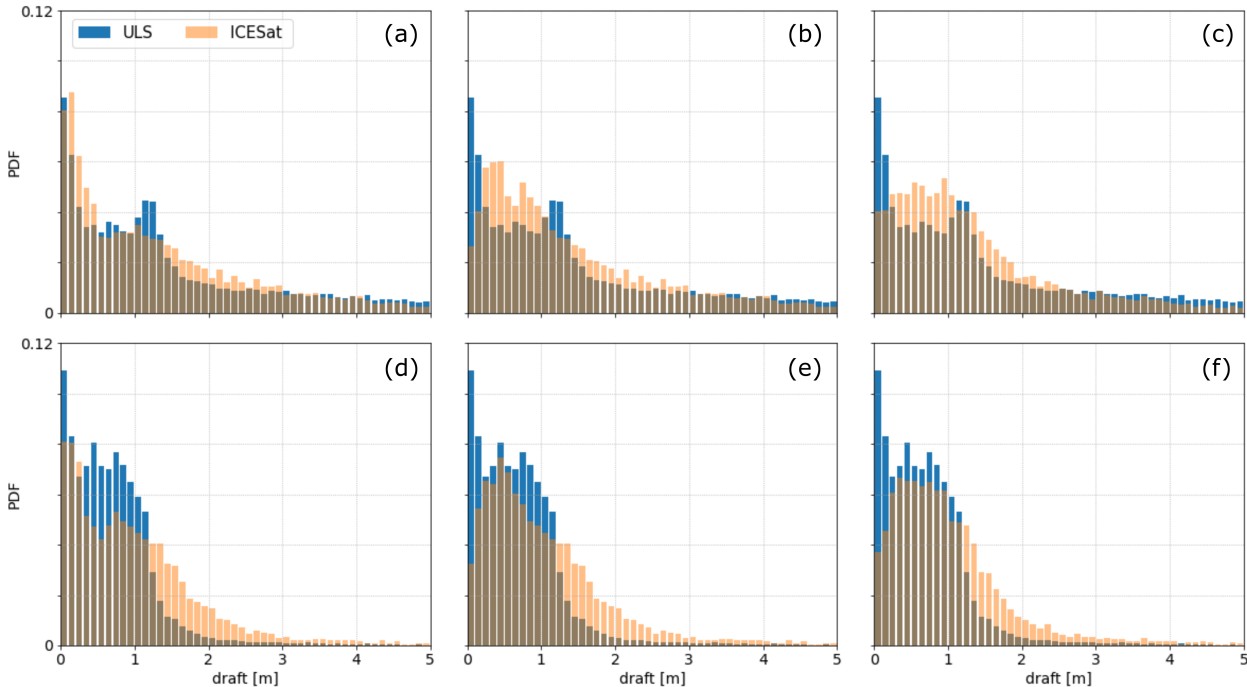

**Figure 11.** Mean sea ice draft distribution from ULS and ICESat measurements. (a,b,c) Western Davis Strait, (d,e,f) Eastern Davis Strait. (a,d) PMW snow depth, (b,e) PMW snow depth with sigmoidal redistribution, (c,f) PMW snow depth with piecewise redistribution. All figures with SnowModel-LG snow density.

significantly different results in March mean sea ice thickness (ranging by ~13%), the spatial pattern of sea ice thickness, and trends in sea ice thickness (ranging from -23 cm/dec to +17 cm/dec). Comparisons with ULS sea ice draft observations demonstrate that snow depth redistribution enables satellite altimetry to capture multiple thickness modes of thin sea ice (<2 m in thickness). Decisions on which snow depth product to use or whether to use a redistribution function can influence the results and conclusions on physical mechanisms driving changes in the ice.

Having identified more consistent sea ice thickness distributions and magnitudes for the two years of CryoSat-2 and ICESat-2 overlap, it is clear that mission overlaps are vital for ensuring long-term SIT trends are robust. None of the used snow depth products provide a good reconciliation between CryoSat-2 and ICESat-2 in mean sea ice thickness nor spatial variability. This shows that more observations on snow depth and density in the seasonal ice zone are necessary to create and validate a suitable snow depth product for this region.

*Data availability.* ICESat surface elevation (http://nsidc.org/data/icesat/), ICESat-2 freeboard (https://nsidc.org/data/icesat-2/data-sets), Operation IceBridge airborne surveys (https://nsidc.org/data/idcsi4), and DMSP SSM/I-SSMIS brightness temperature data (https://nsidc.org/

data/NSIDC-0032/versions/2) are available via National Snow and Ice Data Center (NSIDC). Envisat and Cryosat-2 freeboard data are available from ESA Climate Change Initiative (http://cci.esa.int/seaice). Ice charts are available from the Canadian Ice Service (https://iceweb1.cis.ec.gc.ca/Archive/page1.xhtml?lang=en). ULS draft observations in Davis Strait from the Applied Physics Laboratory, University of Washington are available on http://psc.apl.uw.edu/sea_ice_cdr/Sources/Davis_Strait.html. The merging of CryoSat-2 and SMOS data was funded by the ESA project SMOS & CryoSat-2 Sea Ice Data Product Processing and Dissemination Service and data from March 2011 to March 2020 were obtained from https://www.meereisportal.de (grant: REKLIM-2013-04). The sea ice thickness dataset generated in this study is available on https://data.bris.ac.uk/data/dataset/2peywz756l8182cpwlanm4ratz.

*Author contributions.* IAG and JCL conceptualised the study. IAG carried out the main analysis and wrote the paper. JCL, AAP and NTK contributed to the interpretation of the results. AAP and NTK helped develop the methodology. All authors contributed to revising the manuscript. JCS provided SnowModel-LG data and AAP provided NESOSIM data.

*Competing interests.* The authors declare no competing interests.

*Acknowledgements.* This work was funded primarily by an internal University of Bristol PGR Scholarship of IAG. JCL acknowledges support from the Natural Environment Research Council Project "PRE-MELT" under Grant NE/T000546/1, and the "Diatom-ARCTIC" project (NE/R012849/1) part of the Changing Arctic Ocean programme, jointly funded by the UKRI NERC and the German Federal Ministry of Education and Research (BMBF). JCL received support from the European Space Agency tender "EXPRO+ Snow" under grant ESA AO/1-10061/19/I-EF. JCL acknowledges support from the Centre for Integrated Remote Sensing and Forecasting for Arctic Operations (CIRFA) project through the Research Council of Norway (RCN) under Grant #237906. AAP and NTK were supported by NASA's ICESat-2 Project Science Office. JCS acknowledges her Canada 150 Chair and funding from NASA under grant NASA 15-CRYO2015-0019/NNX16AK85G15.

**References**

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
