# Peer review of "Impacts of snow data and processing methods on the interpretation of long-term changes in Baffin Bay early spring sea ice thickness"

_The Cryosphere, 2021_

## Author Response (AR1)

We thank the anonymous referee for their quick and useful comments. We appreciate the time and effort dedicated to providing feedback on our manuscript and are grateful for the comments. We believe we have been able to address each of the comments.

*L314: "different for different" suggest to change to "vary for different"*

Thank you for this suggestion. We have made this change.

*Figure 6/8: While from what I understood, the CryoSat-2 CCI product mainly plays a role in the supplementary data, and I could probably guess correctly which uncertainty envelope belongs to which data set it can not directly be identified from the Figure or the legend as at least in my print out, they look exactly the same. I would encourage the authors to change this in some way or at least add a hint in the figure caption for clarification.*

Thank you for pointing this out. We have changed the colours of the two CryoSat-2 envelopes to make them more distinct. We hope this will make it easier to distinguish the two.

*L354 and L379: In several occasions, I find a change between "east-west" and "west-east" asymmetry in sea ice thickness in the manuscript. Now I am unsure whether I missed something or this is the lack of me being not a native speaker but this appears inconsistent to me and I suggest to change this.*

We have changed this to make consistently call it 'west-east' asymmetry.

*L366: I assume this refers to the CS-2 LARM product? Please clarify.*

Yes, this is the CS-2 LARM product, we have clarified this in the text.

*Figure 9c: I find the colormap choice not ideal with white at the high end and the general range up to 25 weeks. The resulting maps already appears to be rather "step"-ish, hence, I would suggest to use either something more qualitative or limit the number of classes depending on the exact values to week ranges for an overall better readability.*

We have made the colormap qualitative and gave the high end a light blue colour instead of white.

*L416: Same as comment for L366.*

This is true for both the CS2-LARM and the CS2-CCI product. We have clarified this in the text.

**Anonymous Referee #2**

We thank the anonymous referee for their quick and useful comments. We appreciate the time and effort dedicated to providing feedback on our manuscript and are grateful for the comments and believe we have been able to address each of them.

1. *The long-term sea ice variation in this study is limited to March (early spring) rather than the freezing season or all year round. So, to avoid misunderstanding, I would suggest the study period, i.e., March (early spring) should be addressed in the title of this study.*

   As suggested by the reviewer, we have added 'early spring' to the title.

2. *I really understand that the field observations in Baffin Bay is rather limited, but some ULS-based SIT observations would be helpful. For example, some information can be obtained from Curry et al. (2014, https://doi.org/10.1175/JPO-D-13-0177.1) or Davis Strait Freshwater Flux Array.*

   Thank you for this suggestion. We have added a comparison of ICESat to ULS draft observations (2006-2008, as available on http://psc.apl.uw.edu/sea_ice_cdr/Sources/Davis_Strait.html) in Davis Strait from Curry et al. (2014). We include a comparison of the mean and distribution of drafts around the buoy locations with the ICESat along-track observations for all processing methods in the Supplementary Materials (S5). We have added the main results of this to the main paper in a new section in the results which is further discussed in section 4.3.

3. *Adding some comparisons (e.g., Figure 5) with the AWI CS2SMOS SIT would be interesting because the present AWI retrieval product (CryoSat-2) used AMSR-2 snow depth climatology in Baffin Bay.*

   We have added a comparison of the AWI CS2/SMOS SIT product with the CryoSat-2 LARM SIT product in the Supplementary Materials (S5) which is discussed in the main text in section 4.2. We find thicker sea ice thickness in the CryoSat-2 SIT product than the AWI CS2/SMOS SIT product, which is expected because the SMOS sensor can measure thinner sea ice. The spatial pattern of SIT is similar for both products.

4. *Another suggestion is to add some comparisons with the numerical model results (e.g., PIOMAS), although the CIS charts has been already used. I would also suggest to show some comparison with an ensemble based estimation of the sea-ice variations in the Baffin Bay (Min et al., 2021, https://doi.org/10.5194/tc-15-169-2021).*

   We have briefly looked into a comparison with PIOMAS. The PIOMAS sea ice thickness product seems to show very little spatial variability in Baffin Bay. We have not added this to the manuscript as we believe this will raise more questions than it gives answers to the research questions. We have added a comparison with the ensemble-based estimation of SIT from Min et al. (2021) in the Supplementary Materials (S5) which is discussed in the main text in section 4.2. We also find thicker sea ice from the mean of all processing methods with CryoSat-2 than from the Min et al. (2021) ensemble-based estimation in most of Baffin Bay.

5. *The main findings in this study are significant, however, some deeper explanations/discussions on how the different snow depths influence the different SIT retrievals is also interesting.*

We have added a discussion on how a potential bias of snow depth affects the SIT retrievals for the different sensors in section 4.2. This discusses how a potential negative bias in a snow depth product leads to an overestimation in SIT from laser altimetry and an underestimation in SIT from radar altimetry, leading to an underestimation (or more negative) trend in sea ice thickness when combining ICESat and CryoSat-2 in estimating a trend.

**List of relevant changes**

- Added a comparison of ICESat sea ice thickness measurements with ULS draft observations from a mooring array (Curry et al. 2014) to the discussion (section 4.4) to include a comparison with in-situ data.
- Added a comparison with the AWI CS2SMOS SIT product to the Supplementary Materials, shortly discussed in the main text (section 4.2).
- Added a comparison with the ensemble-based product in Min et al. (2021) in the Supplementary Materials, which is shortly discussed in the main text (section 4.2).
- Added a discussion on how a potential bias of snow depth affects the SIT retrievals for different sensors in section 4.2.
- Added clarifications and removed typos.

---

## Author Response (AR2)

Equation 2: The second term should be positive (e.g. see equation 2 of Price et al. 2019 http://www.the-cryosphere.net/13/1409/2019/). Please confirm whether this is a typo or if this form of the equation was used in the analysis.

*I would like to thank the editor very much for catching this typo! I can confirm this minus sign is not used in the analysis.*

Line 260 - I could not find this parameterisation for the speed of light correction in the reference given. However, Ulaby usually refers to snow in units of g / cm^3. Please could you check this equation, add the equation number to the reference and if necessary correct the equation for the units used in your paper.

*Yes, the editor is right. The equation I included in the paper is in snow units of g/cm^3, I've changed this to kg/m^3, as was also used in the analysis. I made a mistake in entering the reference. The equation is not found in Ulaby Microwave Remote Sensing volume II (1982) but in volume II (1986). I have changed this in the paper. The equation number is E.80: $\epsilon'_{ds} = (1 + 0.51\rho_s)^3$. This gives the real part of the dielectric constant of snow. As the speed of light through a medium is $v = \frac{c}{\sqrt{\epsilon_r}}$ , this leads to the equation in the paper. The equation number has been added to the reference list. As this equation is often used in radar altimetry studies of sea ice (Mallett et al., 2020; Landy et al., 2017; Kwok & Markus, 2018; Kwok, 2014; Kacimi & Kwok, 2020), we believe we do not need to go into depth in the manuscript itself on how this equation is derived.*

Line 296 - 'very little' -> few

*We have made this change.*

Line 449-451, particularly 'the assumed snow layer would weight down and suppress the ice underwater more than the actual layer does' argument is a little hard for me to follow: no more mass is added and in fact less of the column is assumed to be ice. Perhaps it would be better to refer to equation 2 - with greater assumed snow depth (h_s) the first term would increase due to the snow propagation correction (equation 3) and the second term (with correct sign) would also increase due to the adjustment of hydrostatic balance (compared with equation 1: increase in snow depth decreases h_i).

*I have extended this explanation and also included a mathematical explanation referring to the equations like the editor suggested. I personally like to understand things physically, so I've kept the explanations in physical terms as well.*

Supplement page 3 and 5: switch east-west to west-east for consistency

*We have made this change.*

Otherwise I think the inclusion of the ULS data is very interesting. Please spell out the acryonym and perhaps briefly explain the difference between sea ice thickness and draft.

*We have included this.*